# You Only Query Once: An Efficient Label-Only Membership Inference Attack

**Yutong Wu[1], Han Qiu[2]\*, Shangwei Guo[3], Jiwei Li[4,5], Tianwei Zhang[1]**

[1]Nanyang Technological University, [2]Tsinghua University, [3]Chongqing University,
[4]Zhejiang University, [5]Shannon Ai

`yutong002@e.ntu.edu.sg, qiuhan@tsinghua.edu.cn, swguo@cqu.edu.cn`
`jiwei_li@shannonai.com, tianwei.zhang@ntu.edu.sg`

## Abstract

As one of the most severe privacy threats to machine learning models, the membership inference attack (MIA) tries to infer whether a given sample is in the original training set of a victim model by analyzing its outputs. Recent studies only use the predicted hard labels to achieve impressive membership inference accuracy. However, such label-only MIA approach requires very high query budgets to evaluate the distance of the target sample from the victim model's decision boundary. We propose YOQO, a novel label-only attack to overcome the above limitation. YOQO aims at identifying a special area (called *improvement area*) around the target sample and crafting a query sample, whose hard label from the victim model can reliably reflect the target sample's membership. YOQO can successfully reduce the query budget from more than **1,000**× to only **ONCE**. Experiments demonstrate that YOQO is not only as effective as SOTA attack methods, but also performs comparably or even more robustly against many sophisticated defenses. Our code is available at https://github.com/WU-YU-TONG/YOQO.

# 1 Introduction

Recent years have witnessed the widespread applications of machine learning in our daily life. Training a high-quality model requires massive data, which may contain sensitive information. Prior studies (Choquette-Choo et al., 2021; Li & Zhang, 2021; Long et al., 2018; Salem et al., 2018; Shokri et al., 2017) have proven that machine learning models are capable of memorizing most of the training data, leading to the possibility of membership inference attacks (MIAs), where an adversary can infer whether a target data point is in the training set of the victim mode or not.

Most of the existing works (Carlini et al., 2022; Liu et al., 2022; Long et al., 2018; Salem et al., 2018; Shokri et al., 2017; Yuan & Zhang, 2022) exploit loss functions or prediction scores to conduct MIAs. The adversary feeds the target sample to the victim model and obtains the outputs in the form of posteriors, which are subsequently processed by a binary classifier to make membership decisions. Therefore, one effective defense against these attacks is to simply mask the posteriors (Jia et al., 2019) or only return hard labels.

To break these defenses, researchers further proposed the label-only MIA (e.g., boundary attack (Choquette-Choo et al., 2021; Li & Zhang, 2021)), which requires only hard labels to infer the membership of the target sample. This attack estimates the distance between the target sample and its nearest decision boundaries by querying the victim model iteratively, and then decides the membership by categorizing samples with longer distances as members and shorter distances as non-members. It has brought MIAs to a much more practical scenario, as many Machine Learning as a Service (MLaaS) platforms only return hard labels to users (Foundation, 2021).

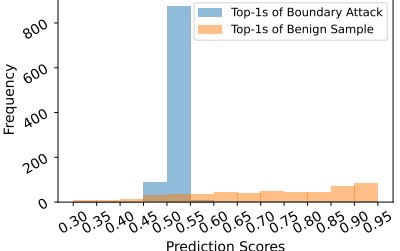

Figure 1: Distributions of top-1 scores of benign queries and malicious queries.

---

\*Corresponding Author

However, the boundary attack requires a high query budget, usually more than $1000\times$ queries to check each sample (Choquette-Choo et al., 2021; Li & Zhang, 2021). This leads to the following limitations. (1) The attack is rather costly given that MLaaS normally charges users based on the number of queries. (2) It is easier to detect this attack, due to the large number of anomalous queries. Fig. 1 compares the distribution of the top-1 scores for a victim model under the boundary attack and benign environment. Their distinct difference indicates that iterative queries put the boundary attack in jeopardy of being detected and disturbed by statistics-based defenses like PRADA (Juuti et al., 2019). (3) As the boundary attack requires accurate distance estimation, Rajabi et al. (2022) proposed a defense called LDL against it. LDL creates a hyper-sphere around the samples in which the decisions of the victim model will not alter. It successfully enervates the boundary attack as the gap attack (Yeom et al., 2018), which has much lower inference accuracy.

To overcome the above limitations of the boundary attack (Choquette-Choo et al., 2021; Li & Zhang, 2021), we propose YOQO ("**Y**ou **O**nly **Q**uery **O**nce"), a novel label-only MIA that only requires to query the victim model once to identify the membership of each target sample. The key insight of YOQO lies in the concept of the *improvement area*, which can precisely reflect the contribution of the target sample to the victim model's performance. Any sample inside the improvement area can be used as the query sample to disclose the target sample's membership. Based on this insight, we further propose two approaches (online and offline attacks), which can generate such sample with different efficiency and accuracy trade-offs.

We perform extensive experiments over different datasets (CIFAR10, GTSRB, Location, Texas), and compare YOQO with different label-only MIAs. Evaluations indicate that YOQO can achieve comparable inference accuracy to state-of-the-art attacks, with a much smaller query budget (once). We also show that YOQO is more robust against various defense solutions to existing attacks.

## 2 PRELIMINARY

### 2.1 MEMBERSHIP INFERENCE ATTACK

MIA (Shokri et al., 2017) has drawn great attention since it reflects the memorization ability of machine learning models over their sensitive training data. The adversary tries to infer if a target sample belongs to the training set $\mathcal{D}$ of a victim model $f_{\mathcal{D}}$. He solves this binary classification problem by building an algorithm $\mathcal{A}$ for an arbitrary sample $x$: $\mathcal{A}(f_{\mathcal{D}}, x) = \mathbb{I}(x \in \mathcal{D}))$, where $\mathbb{I}(X)$ is the indicator function that equals 1 if $X$ is true and 0 otherwise. There are two main categories of MIA approaches based on the adversary's knowledge.

**Posterior-based MIA**. Numerous works (Carlini et al., 2022; Hayes et al., 2017; Hu & Pang, 2023; Liu et al., 2022; Sablayrolles et al., 2019; Salem et al., 2018; Shokri et al., 2017; Song et al., 2019; Yeom et al., 2018; Yuan & Zhang, 2022; Wen et al., 2022) exploit posteriors (e.g., prediction scores or output logits) to infer the membership of the target sample. There are two strategies to build the algorithm $\mathcal{A}$. The first strategy is to implement a DNN-based binary classifier (Hayes et al., 2017; Liu et al., 2022; Shokri et al., 2017; Yuan & Zhang, 2022). The adversary first prepares a set of shadow datasets $\mathcal{D}_s$, which follow the same distribution as the victim model's training set $\mathcal{D}$. Some of the shadow datasets contain the target sample $x$ while others do not. Then he trains a surrogate model $f_s$ on each shadow dataset. Based on these surrogate models, he can construct a membership dataset: $\mathcal{D}_{mem} = \{(f_s(w), \mathbb{I}(w \in \mathcal{D}_s))\}$, The adversary then trains a binary classifier $g$ on $\mathcal{D}_{mem}$. In the attack phase, he queries the victim model with $x$ and retrieves the outputs, which are then fed to $g$ to infer the membership: $\mathcal{A}(f_{\mathcal{D}}, x) = g(f_{\mathcal{D}}(x))$.

The second strategy relies on statistic-based approaches to learn a threshold $\tau$ to divide the manifold into two classes (Carlini et al., 2022; Hu & Pang, 2023; Sablayrolles et al., 2019; Salem et al., 2018; Song et al., 2019; Yeom et al., 2018). Specifically, the adversary also trains surrogate models on the subsets of a shadow dataset. He then uses the training loss or some specially designed loss functions $F$ to gain a score for each output given by $f_{\mathcal{D}}$. The threshold $\tau$ is subsequently learned by studying the correlation between the memberships and the scores. Thereby, for these works: $\mathcal{A}(f_{\mathcal{D}}, x) = \mathbb{I}(F(f_{\mathcal{D}}(x)) > \tau)$. To get the posterior from the victim model, most works directly use the original target samples, while some works like (Wen et al. (2022)) proposed to query the target model with adversarial examples to further enlarge the variance in the logits between the IN and OUT models to achieve better performance. In this paper, we share the similar thoughts to craft a label-level membership-sensitive query sample.

**Label-only MIA**. The above attacks require the adversary to have the posteriors of the query sample, which is not possible in some MLaaS platforms (Foundation, 2021). To address this issue,

researchers proposed label-only attacks, where the adversary only needs the prediction labels. There are mainly three state-of-the-art label-only attacks. (1) *Gap attack* (Yeom et al., 2018) leverages the overfitting phenomenon to infer the membership. The adversary simply takes samples that are misclassified by the victim model as non-members and the rest as members. (2) *Boundary attack* (Choquette-Choo et al., 2021; Li & Zhang, 2021) follows the intuition that non-member samples are closer to the decision boundary of the victim model. The adversary generates an adversarial example from the target sample using decision-based adversarial attack algorithms like (Chen et al., 2020; Li et al., 2020; Bai et al., 2020), and calculates the distance between them, which will be used as the estimation of the distance between the target sample and its nearest decision boundary. He then trains a bunch of surrogate models and casts the attack on them to learn a threshold $\tau$ to distinguish member and non-member samples. (3) *Data augmentation attack* (Choquette-Choo et al., 2021) exploits the observation that machine learning models are likely to overfit augmented samples. The adversary first creates additional data points from the target sample $x$ with different augmentation strategies. Then he feeds all these samples to a shadow model $\hat{h}$ following the target model's architecture and training data distribution. He compares the returned labels $(l_0, ..., l_n)$ with the ground truth $l_{true}$, and uses the comparison results $b_i = \mathbb{I}(l_{true} = (l_i))$ as features to train the algorithm $\mathcal{A}$ for membership inference. This approach requires the strong assumption that the adversary knows the augmentation strategies for training the victim model, which may not be practical in reality.

## 2.2 Defenses Against MIAs

Multiple solutions are proposed to mitigate MIAs, which can be summarized into four categories.
**Fine-tuning the model**. In Adversarial Regularization (ADV) (Nasr et al., 2018), the defender first trains an attack model to infer the membership. Then he fine-tunes the target model by minimizing the loss on the training set while maximizing the classification loss of the attack model. PPB (Yuan & Zhang, 2022) tries to narrow the prediction gaps between members and non-members to make them indistinguishable. It fine-tunes the target model by minimizing both the KL divergence between the ranked output posterior and classification loss on the training set.
**Perturbing the inputs**. LDL (Rajabi et al., 2022) constructs a hyper-sphere around the target sample by adding noise $n_i \sim \mathcal{N}(0, \sigma)$ to make the decisions of the model unchanged for all samples in the sphere. Samples inside the hyper-sphere are fed into the victim model, and all the outputs are merged by average to get the final prediction.
**Modifying the posteriors**. MemGuard (Jia et al., 2019) injects adversarial perturbations to the confidence to mislead the adversary's membership classifier. One-hot encoding (Yang et al., 2020) encodes the prediction confidence into a one-hot code, making it almost impossible to achieve MIAs just based on the posteriors.
**Differential privacy** (Abadi et al., 2016; Zou et al., 2020). DP-SGD is proven to be effective in preventing privacy leakage (Rahimian et al., 2021; Truex et al., 2019; Zou et al., 2020). It modifies the gradient in the training process by clipping and then adding Gaussian noise to it. As the noise follows the distribution $\mathcal{N}(0, \sigma)$, the standard deviation $\delta$ should be in the order of $\Omega(q\sqrt{T \log{(1/\delta)}}/\epsilon)$.

## 3 Methodology

**Threat Model.** We follow the same threat model from (Choquette-Choo et al., 2021; Li & Zhang, 2021). Specifically, we consider a victim model $f_{\mathcal{D}}$ trained from a training set $\mathcal{D}$. It accepts query data from users and only returns the corresponding hard labels. This is a practical setting in many MLaaS platforms (Foundation, 2021). An adversary tries to perform the MIA over $f_{\mathcal{D}}$: for an arbitrary sample $x$, he wants to check whether it belongs to the training set $\mathcal{D}$. The adversary has clue about the tasks of the victim model. We further assume the adversary can collect data samples that follow a similar distribution as the training samples of the victim model. He is also capable of training shadow models by himself. However, he does not have the knowledge of either the network architecture or parameters of the victim model.

### 3.1 Key Insight

To reduce the attack cost and risk of being detected, we further restrict the attack budget to querying $f_{\mathcal{D}}$ once. Therefore, instead of directly sending the target sample $x$ to $f_{\mathcal{D}}$, we aim to craft a new query data $x'$ from $x$, which can better reflect the membership of $x$. Assuming the ground-truth label of $x$ is $l$, then $x'$ should satisfy the following two properties:

- **Specificity**: for any model $f_{out}$ whose training set does not include $x$ (dubbed *out-model*), it predicts a different label for $x'$ from $l$.

- **Sensitivity**: for any model $f_{in}$ whose training set includes $x$ (dubbed *in-model*), it predicts the same label $l$ for $x'$.

With such properties, we can tell the membership of $x$ based on the returned label of $x'$. To identify $x'$, we introduce the concept of *improvement area*, which is the set of all qualified query samples $x'$. This is based on the phenomenon that the performance of a machine learning model on certain classes increases when new samples are added to the training set (Salem et al., 2018; Liu et al., 2022). Basically, each newly added sample can provide special features for the model to learn, which will change the model's decision boundary. Fig. 2 visualizes this process. A new sample (blue rhombus) added to the training set can drive the boundary farther from it locally, resulting in a green zone, which is the improvement area. All the data points within the improvement area are originally assigned to the orange class and switched to the blue class due to the new sample. Therefore, the adversary's goal is to discover a data point from this area to query the remote model for membership inference.

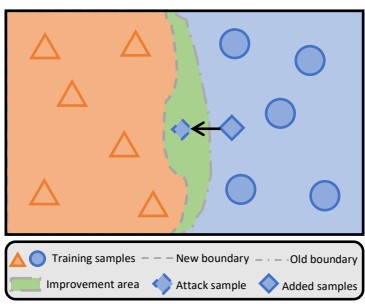

Figure 2: Illustration of improvement area and query samples.

Although samples in the improvement area can effectively disclose the membership of the target sample $x$, it is non-trivial to identify such area. When only $x$ is added to the training set, its influence on the prediction accuracy is minor, leading to a very small improvement area. Below we propose two approaches to generate query samples.

## 3.2 ONLINE ATTACK

The adversary first prepares $N$ training sets, denoted as $\mathcal{D}_{out}^i$ for $i \in [1, N]$. These datasets are composed of data samples whose membership is not interesting to the adversary. He then trains $N$ out-models from these training sets: $f_{out}^i$. These out-models can be consistently used to check the membership status of all other samples.

For a target sample $x$ with the ground-truth label $l$, to generate the corresponding query sample $x'$ that satisfies the specificity property, the adversary needs to enforce $l \neq f_{out}^i(x')$ for $i \in [1, N]$. This can be realized with the following optimization objective, where $CE$ stands for the cross entropy:

$$x' = \arg\max_a \sum_{i=1}^{N} CE(f_{out}^i(a), l) \tag{1}$$

To meet the sensitivity property for $x'$, the adversary needs to construct $N$ training sets $\mathcal{D}_{in}^i = \mathcal{D}_{out}^i \cup \{x\}$, and trains the corresponding in-models $f_{in}^i$. Then he expects $x'$ to be classified as the ground-truth label $l$, which is converted to the following objective:

$$x' = \arg\min_a \sum_{i=1}^{N} CE(f_{in}^i(a), l) \tag{2}$$

Based on the two requirements, we formulate the following loss function for optimizing $x'$:

$$x' = \arg\min_a \sum_{i=1}^{N} \left( CE(f_{in}^i(a), l) - CE(f_{out}^i(a), l) \right) = \arg\min_a \sum_{i=1}^{N} CE\left( \frac{f_{in}^i(a)}{f_{out}^i(a)}, l \right) \tag{3}$$

However, since scalars in both $f_{in}^i(a)$ and $f_{out}^i(a)$ are in $[0, 1]$, the optimization process is lopsided towards the specificity term (Eq. 1) regardless of the sensitivity property (Eq. 2). To address this issue, we can covert Eq. 1 to a minimization problem. Specifically, we choose $l_i'$, which is the predicted label of $f_{out}^i(x)$ with the highest confidence score other than the ground-truth label $l$: $l_i' = \arg\max_a f_{out}^i(x)_a \quad s.t.\ a \neq l$. Then we aim to achieve $l_i' = f_{out}^i(x')$ for $i \in [1, N]$. We choose the label $l'$ because the generated $x'$ will be closer to $x$. Since the decision boundary closer to $x$ is more probable to be changed by the addition of $x$, it is also easier to find qualified query samples whose labels predicted by the out-model $f_{out}^i$ is $l'$. The optimization objective is

$$x' = \arg\min_a \Big( \alpha \cdot \sum_{i=1}^{N} CE(f_{out}^i(a), l_i') + \sum_{i=1}^{N} CE(f_{in}^i(a), l) \Big) \tag{4}$$

where $\alpha$ is a hyper-parameter to balance the two terms. Algorithm 1 describes the details of our online attack. In our implementation, we use stochastic gradient descent to generate $x'$.

---

**Algorithm 1:** Online attack

**input** : target sample $x$ and corresponding ground-truth label $l$, victim model $f$, $N$ out-dataset $\mathcal{D}_{out}^i$ and corresponding out-models $f_{out}^i$

**output:** membership of $x$

1 **for** $i := 1$ *to* $N$ **do**
2     $\mathcal{D}_{in}^i \leftarrow \mathcal{D}_{out}^i \cup \{x\}$
3     $f_{in}^i \leftarrow$ TRAINWITH($\mathcal{D}_{in}^i$)                          ▷ Shadow training
4     $l_i' \leftarrow \arg\max_k f_{out}^i(x)_k, \; s.t. \; l_i \neq l$
5 **end**
6 Generate $x'$ by optimizing Eq. 4
7 $l' \leftarrow f(x')$                                               ▷ Querying victim model
8 **if** $l' = l$ **then**
9     **return** *'Member'*
10 **else**
11     **return** *'Non-member'*
12 **end**

---

### 3.3 OFFLINE ATTACK

In our online attack, the $N$ out-datasets $\mathcal{D}_{out}^i$ and out-models $f_{out}^i$ can be reused for any data sample not in these datasets. However, for each target sample $x$, the adversary has to generate the corresponding in-datasets $\mathcal{D}_{in}^i$ and in-models $f_{in}^i$, which is costly for training models. To address the efficiency issue, we propose an offline attack which only requires the out-datasets and out-models to infer the membership. This is inspired by (Elliott et al., 2021) which generates a counterfactual explanation for a sample $x$.

We still use the out-models to enforce the specificity property. For sensitivity, instead of using the in-models, we adopt a simple normalization term to restrict the distance between $x$ and $x'$. Then the generated $x'$ has a higher chance to be assigned with the same label as $x$ by any in-model. The final optimization goal is as follows:

$$x' = \arg\min_a \Big( \sum_{i=1}^{N} CE(f_{out}^i(a), l_i') + \gamma \cdot MSE(a, x) \Big) \tag{5}$$

where $MSE(\cdot, \cdot)$ stands for mean square error, and $\gamma$ is a hyper-parameter to balance the two terms. The adversary only needs to remove the operations in Lines 2 and 3 in Algorithm 1 (which are costly), and replace Eq. 4 with Eq. 5 in Line 6 to conduct the efficient offline attack.

## 4 EVALUATION

We evaluate our online and offline attacks under various settings to prove their effectiveness and efficiency. We compare our attacks with SOTA MIAs to show our superiority.

**Datasets**. YOQO is general to different tasks. Without loss of generality, we follow (Choquette-Choo et al., 2021; Carlini et al., 2022; Li & Zhang, 2021) to test YOQO on several classical visual tasks such as CIFAR-10, CIFAR-100 (Krizhevsky et al., 2009), GTSRB, SVHN (Netzer et al., 2011) and Tiny-ImageNet. We also test YOQO on two tabular datasets: Location (Yang et al., 2015) and Texas.

**Model architectures**. Following prior works (Carlini et al., 2022; Choquette-Choo et al., 2021; Li & Zhang, 2021), we use CNN7, ResNet18, ResNet34, DenseNet121, Inceptionv3 and VGG16 for image datasets. For tabular datasets, we follow (Yuan & Zhang, 2022) to use a fully connected network consisting of two hidden layers, with the size of 256 and 128, respectively.

**Baseline attacks**. We consider three label-only MIAs discussed in Section 2.1: gap attack (Yeom et al., 2018), boundary attack (Choquette-Choo et al., 2021; Li & Zhang, 2021), and data augmentation attack (Choquette-Choo et al., 2021). We follow the settings in (Choquette-Choo et al., 2021)

to implement the gap attack and data augmentation attack. For the later, we use the rotation and translation strategies to craft the augmented queries. For the boundary attack, we train 16 pairs of shadow models to select the best thresholds so as to conduct a fair comparison. To craft the shadow models, we randomly split the dataset to be inferred into two halves, and blend one of them with the training set before we train the shadow models. Then we conduct the boundary attack on these models, and determine the best threshold that can achieve the best performance.

Table 1 shows the query budgets of the three label-only attacks and YOQO. We can see YOQO is much more efficient than the boundary attack. The data augmentation attack requires more information about the victim, which is not practical and is beyond the scope of our threat model. The gap attack also just needs 1 query, but the inference accuracy is much lower than YOQO (see Section 4.1).

| Attack | Queries |
|---|---|
| Boundary attack | 20,000+ |
| Data augmentation attack | 10+ |
| Gap attack | 1 |
| **YOQO** | **1** |

Table 1: Query budget comparisons between different label-only MIAs.

**MIA Defenses**. Five state-of-the-art defense strategies are implemented to test the robustness of our attacks: ADV, PPB, DP-SGD, LDL, and MemGuard (Section 2.2). For ADV and PPB, we pre-train the victim model to have up to 64.3% accuracy on the validation set and then fine-tune the model for 50 epochs. For LDL, we set the sample times for each query as 200. We set all the hyper-parameters in these defenses following (Yuan & Zhang, 2022).

**Implementations**. We train all the networks until they achieve more than 98% accuracy on the training set. We use Adam as the optimizer with all the hyper-parameters default in its PyTorch implementation. To conduct the online attack, we train 16 in-out model pairs and set $\alpha = 2$. For the offline attack, we only use 16 out-models with $\gamma = 5$. The gradient descent algorithm is used to optimize $x'$. For the two attacks, we set the stop threshold to 4 and 8, and the maximum number of iterations to 30 and 35 respectively.

**Metrics**. We use the membership inference accuracy as our metric to evaluate the performance of all referred MIAs. It equals the accuracy of the binary classification task that tells non-members from members. We also use Precision and Recall for more detailed investigations. We use 500 data points for all the evaluations, in which 250 samples are members and the other 250 are non-members. The result is mainly the average accuracy of five repeated experiments.

Note that we do not use the metric proposed in (Carlini et al., 2022), which measures the true postive

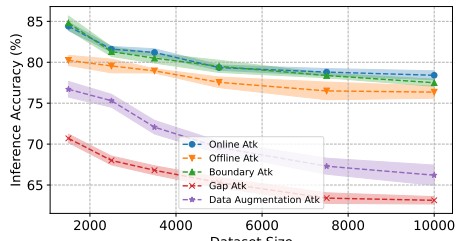

Figure 3: Membership inference accuracy of label-only MIAs on CIFAR-10.

rate (TPR) of MIAs at a low false positive rate (FPR). Unlike other MIAs whose TPR and FPR can be easily controlled by choosing the threshold, YOQO only has one hard label as the output and cannot adjust TPR/FPR, making this metric inapplicable.

## 4.1 ATTACK EFFECTIVENESS

**Comparisons with prior attacks**. Fig. 3 shows the membership inference accuracy of different attacks on CIFAR-10. We use CNN7 as both shadow and victim models and change the size of their training set to see its impact on the attack performance. We observe that our online attack achieves almost the same inference accuracy as the boundary attack, while the latter requires over 20,000 queries per sample. Despite the offline attack being less powerful, it still surpasses the gap attack by approximately 10% in terms of inference accuracy. Our two attacks can also beat the data augmentation attack. We also notice that the inference accuracy of all the attacks dwindles when the size of victim model's training set increases. This is because the model trained on a bigger training set tends to be less overfitting, and has less membership privacy leakage.

**Impact of model architectures**. We test the performance of YOQO on different architectures for the shadow model and victim model, and the results are in Fig. 4. All the victim models are trained on subsets consisting of 2,500 training data from CIFAR-10. Here "Assembly" means to use the models of all the tested architectures, including CNN7, VGG18, ResNet18, DenseNet121, Inceptionv3, and SeResNet18, to form a model ensemble for the generation of the query sample $x'$. We train four

in-out pairs for each architecture, leading to a total of 40 models in the ensemble (20 in-models and 20 out-models). Intuitively, the membership inference accuracy should be the highest when the shadow model is of the same structure as the victim one. However, as the performance of different model architectures is diverse, their overfitting degree on the datasets varies. This gives the counter-intuitive results in Fig. 4: the inference accuracy of some architectures is constantly higher than others. For example, the inference accuracy of SeResNet18 is higher than other models in most cases, while the accuracy of DenseNet121 is always the lowest. Table 2 further compares the inference accuracy of YOQO with the gap attack and boundary attack ($l_0$, $l_1$, $l_2$, $l_\infty$ stand for different norms to represent the distance). The training set of the victim model contains 2,500 training samples and the shadow models are trained on the shadow training set of the same size. The results show that our attacks tend to have stably higher attack accuracy than the other two attacks.

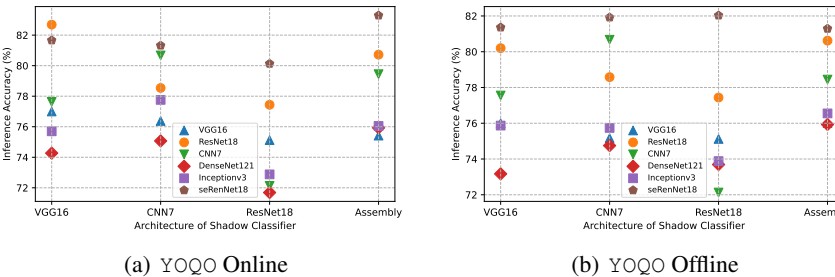

(a) YOQO Online         (b) YOQO Offline

Figure 4: Membership inference accuracy of YOQO on CIFAR-10 against different victim models.

| Victim Model | CNN7 | VGG16 | ResNet18 | DenseNet121 | Inceptionv3 | SeResNet18 |
|---|---|---|---|---|---|---|
| Gap Attack | 67.85 | 62.7 | 72.71 | 66.50 | 60.15 | 72.2 |
| Boundary ($l_0$) | 69.56 | 73.06 | 74.25 | 68.13 | 71.65 | 73.06 |
| Boundary ($l_1$) | 81.25 | 63.81 | 83.38 | 68.14 | 60.75 | 74.37 |
| Boundary ($l_2$) | 81.38 | 63.82 | 83.88 | 68.19 | 61.06 | 76.50 |
| Boundary ($l_\infty$) | 81.50 | 67.44 | **83.88** | 73.19 | 68.12 | **86.00** |
| YOQO Online | **81.65** | 76.20 | 78.50 | **75.00** | 77.56 | 81.87 |
| YOQO Offline | 79.19 | **76.69** | 79.31 | 73.75 | 75.06 | 81.68 |

Table 2: Membership inference accuracy of MIAs with various model structures. The experiment is conducted on CNN7 models with the CIFAR-10 dataset of 2,500 samples.

**Attack effectiveness on various tasks**. We examine the adaptability of our attacks to different tasks and datasets. Table 3 shows the results. Generally, our attacks are more effective than the gap attack on all the investigated tasks. However, YOQO performs relatively poorer on tabular data than image ones. We hypothesize it is because the tabular data are of much lower dimensions than the images. While YOQO is able to generate more fine-grained membership query data on visual tasks, the tabular data may have more restrictions on it, leading to worse results. Another possible reason is that the model architectures may also have an impact on privacy leakage, making our methods less suitable for full-connected neural networks.

| Task Name | CIFAR10 | CIFAR100 | GTSRB | SVHN | Tiny-ImageNet* | Tiny-ImageNet | Location | Purchase100 |
|---|---|---|---|---|---|---|---|---|
| Model Arch | CNN7 | CNN7 | CNN7 | CNN7 | ResNet34 | ResNet34 | ColumnFC | ColumnFC |
| Size of Training Set | 2,500 | 9,000 | 1,000 | 3,500 | 75,000 | 75,000 | 1,000 | 6,000 |
| Boundary Attack | 81.38 | 87.59 | 59.90 | 71.83 | 73.21 | 81.03 | 82.27 | 74.17 |
| Gap Attack | 67.85 | 79.75 | 54.12 | 63.09 | 65.20 | 75.70 | 70.94 | 63.89 |
| YOQO Online | 81.65 | 86.10 | 67.16 | 73.94 | 73.15 | 80.94 | 77.50 | 68.28 |
| YOQO Offline | 79.19 | 84.75 | 67.15 | 71.75 | 70.47 | 77.65 | 83.25 | 70.00 |

Table 3: Membership inference accuracy of MIAs on various tasks. '*' means the models are pre-trained on ImageNet.

**Conclusion**. From the above experiments, we conclude that YOQO is as effective as boundary attack and much better than data augmentation attack on models with different overfitting levels. It only requires to query once and does not need extra information about victim's training details. YOQO is also adaptive to different tasks and network architectures for both the victim and shadow models.

## 4.2 ABLATION STUDY

We perform ablation studies to disclose the influence of different factors. We use CIFAR10 for all the evaluations and train CNN7 as shadow models to generate membership query samples. The size of the training datasets is 2,500 by default.

**Size of the shadow training sets**. Fig. 5 shows the attack results with different sizes of the victim model's training set ($x$-axis) and shadow training set ($y$-axis). Intuitively, we expect to find the best performance on the diagonal, as the adversary has more information about the victim model. For the online attack (Fig. 5(a)), the results are aligned with this intuition when the victim models are trained on datasets of 1,500, 5,000, and 10,000 samples. Yet the facts contradict when it comes to 2,500, 3,500, and 7,500 samples. This shows the size of the shadow training set has little impact on the online attack performance. For the offline attack (Fig. 5(b)), generally the left bottom corner has higher accuracy than the right top, revealing that the offline attack performs better when the shadow models are trained on bigger shadow sets. This is because the adversary can only generate the query sample $x'$ according to the gradient of the out-models, which cannot guarantee the sensitivity requirement. When the training set is smaller, it is likely to be more unbalanced, resulting in a more inaccurate direction for the adversary to optimize $x'$, so the attack performance is degraded.

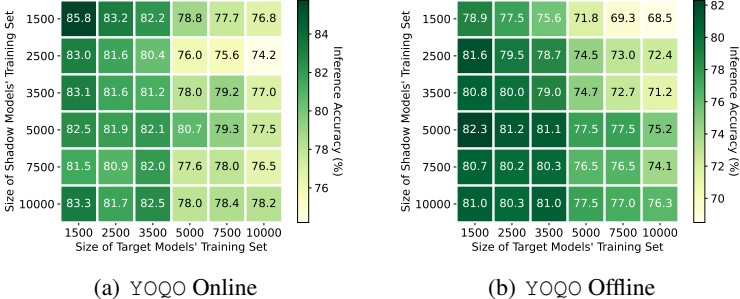

(a) YOQO Online  (b) YOQO Offline

Figure 5: Membership inference accuracy of YOQO with varied sizes of shadow/victim training sets.

**Hyper-parameter in loss functions**. We now investigate the hyper-parameters in the attack's loss functions, i.e., $\alpha$ in Eq. 4 and $\gamma$ in Eq. 5, which scale the specificity and sensitivity terms. Fig. 6(a) shows the online attack performance trend with $\alpha$. With a larger $\alpha$, the precision increases as there are fewer false positive samples, whereas the recall decreases because the influence of the sensitivity term is enervated, leading to more false negatives. Fig. 6(b) shows the offline attack performance. When increasing $\gamma$, the recall increases and the precision decreases, indicating more false negatives and fewer true positives. However, both $\alpha$ and $\gamma$ have a small influence on the inference accuracy when they fall in most parts of the given intervals.

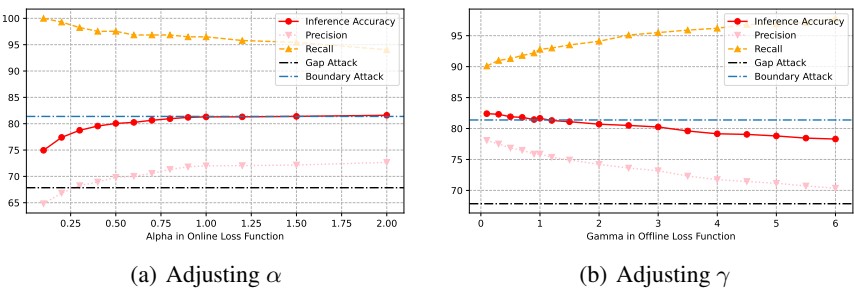

(a) Adjusting $\alpha$  (b) Adjusting $\gamma$

Figure 6: Performance of YOQO as $\alpha$/$\gamma$ changes.

**Number of in-out model pairs**. We use different numbers of in-out pairs to generate the query sample $x'$, and the corresponding inference accuracy is shown in Fig. 7. The accuracy of both online and offline attacks increase when we use more models in the generation process. When there are more in-out pairs, $x'$ can better meet the two requirements in Section 3.1, which leads to stronger transferability according to (Liu et al., 2016). Note that the attack accuracy increases marginally with more than 16 in-out model pairs. So we set this hyper-parameter as 16 to trade-off the efficiency and performance.

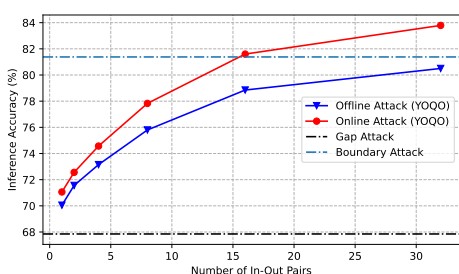

Figure 7: Membership inference accuracy of YOQO versus the number of in-out pairs.

**Selection of $l_i'$**. In Section 3.2, we select the nearest class as $l_i'$ for the specificity term. Table 4 compares

the attack performance with this selection, and randomly choosing a label. The attacks are evaluated over CIFAR10. It is obvious that selecting the nearest class can better facilitate the attack than selecting a random label for $l_i'$. This is aligned with the hypothesis in (Nasr et al., 2018). As the models tend to give relatively high and stable prediction scores to members (Liu et al., 2022; Yeom et al., 2018; Yuan & Zhang, 2022), a significant impact of converting a non-member sample to the member is the plummet of the second biggest prediction score, as the prediction logits are scaled by the softmax layer and always sum up to 1. In other words, the decision boundary is mostly changed between the class with the second biggest confidence and the ground-truth class. The boundary of a random class other than that may remain the same, leading to a smaller improvement area. This makes it harder to search for a query sample.

## 4.3 ROBUSTNESS AGAINST DIFFERENT DEFENSES

We evaluate the robustness of YOQO against various state-of-the-art MIA defenses. All the attacks are conducted on CIFAR10 and CNN7, with 2,500 samples in the training set. Fig. 8 presents the victim model's clean accuracy ($y$-axix) and adversary's inference accuracy ($x$-axis) when we vary the hyperparameter values in these defenses. Ideally, a good defense can reduce the inference accuracy while pre-

| Attacks | Choose randomly | Choose the nearest |
|---|---|---|
| Gap Attack | 67.85 | 67.85 |
| YOQO Online | 75.15 | 81.65 |
| YOQO Offline | 73.42 | 79.19 |

Table 4: Membership inference accuracy of YOQO using different ways to choose $l_i'$.

serving the model's clean accuracy. We observe that some defenses (e.g., PPB and ADV) can have effects on our attacks, but cannot satisfactorily mitigate them while keeping the functionality of the victim model simultaneously. Other defenses like MemGuard have negligible effects on YOQO, as they are mainly targeting the posterior-based attacks.

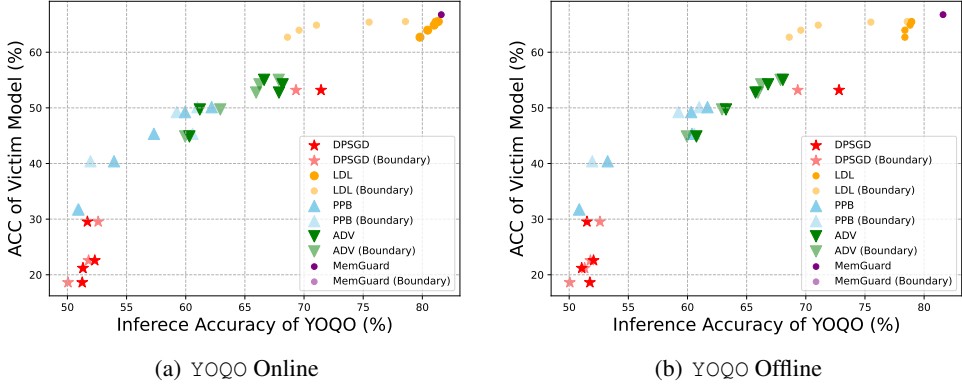

(a) YOQO Online

(b) YOQO Offline

Figure 8: Membership inference accuracy against different defenses.

LDL is a defense solution specifically designed for label-only MIAs. From the evaluation, we observe that LDL can significantly reduce the inference accuracy of the boundary and gap attacks in most cases. In contrast, the accuracy of our two YOQO attacks remain high and stable. We hypothesize that this is because YOQO is less dependent on the concrete distances than the other attacks, so LDL can hardly influence the improvement area.

## 5 CONCLUSION

We propose YOQO, a novel label-only membership inference attack to reduce the heavy query budget. We demonstrate that privacy leakage can still take place when the adversary is allowed to query the model just once. We introduce the concept of improvement area to analyze the label-only MIAs and design two novel techniques to effectively generate the query sample and accurately make the membership decision. Evaluations demonstrate that YOQO can achieve comparable inference accuracy as the state-of-the-art boundary attack, which requires more than $1000\times$ queries. Besides, YOQO is more robust against different MIA defenses compared to the boundary attack.

## 6 ACKNOWLEDGEMENT

Thi work is supported by National Natural Science Foundation of China No. 62106127 and the National Research Foundation, Singapore, the Cyber Security Agency under its National Cybersecurity R&D Programme (NCRP25-P04-TAICeN), the Singapore Ministry of Education (MOE) AcRF Tier 2 MOE-T2EP20121-0006.

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

# A APPENDIX

## A.1 THE TRAINING RECIPES OF THE MODELS EVALUATED

In Table A.1 we provide the training recipes of the models tested in § 4. The training settings basically follow such recipe unless specially referred.

| Dataset | Model | Learning Rate | Optimizer | Batch Size | Training Epoch |
|---|---|---|---|---|---|
| CIFAR10/CIFAR100/gtsrb/svhn | CNN7 | 0.001 | Adam | 128 | 30 |
| | VGG16 | 0.001 | Adam | 64 | 50 |
| | ResNet18 | 0.001 | Adam | 64 | 20 |
| | DenseNet121 | 0.001 | Adam | 128 | 30 |
| | InceptionV3 | 0.001 | Adam | 128 | 25 |
| | SeResNet18 | 0.001 | Adam | 128 | 20 |
| Purchase100/location | ColumnFC | 0.001 | Adam | 32 | 30 |

Table A.1: Training recipes for models evaluated.

## A.2 WORST-CASE STUDY

We further perform the worst-case study to show the feasibility of YOQO. Specifically, we use the distance of a sample from the decision boundary yielded by the boundary attack Choquette-Choo et al. (2021) as criterion, and select the samples with 10% longest distance in terms of $l_0, l_1, l_2$, and $l_\infty$ respectively from the test set as the target samples evenly from all the out and in samples. This makes the set of the target samples be composed of 200 samples (100 out samples and 100 in samples). We evaluate the distance of each sample on 5 victim models, and use the average distance as the criterion. The models being tested are CNN7 model trained on CIFAR10 of 2,500 samples. In Table A.2 we show the results:

| Norm | $l_0$ | $l_1$ | $l_2$ | $l_\infty$ |
|---|---|---|---|---|
| Boundary Attack | 52.7% | 54.0% | 54.1% | 53.3% |
| Gap Attack | 50.0% | 50.0% | 50.0% | 50.0% |
| YOQO Online | 65.3% | 61.0% | 61.22% | 61.5% |
| YOQO Offline | 60.0% | 57.7% | 56.5% | 59.5% |

Table A.2: Worst-case study

We observe that our methods tend to have better performance than the boundary attack. This indicates that YOQO can be more effective to infer the membership of the worst-case samples which the boundary attack is hard to deal with.

## A.3 INFLUENCE OF $L_2$ REGULATIONS WHILE TRAINING THE VICTIM MODELS.

To demonstrate the effectiveness of YOQO on less overfitting models, we further perform evaluations on models trained with stronger L2 regulations. Specifically, we use CNN7 as both the shadow and victim model, over 2,500 CIFAR-10 images as the training set. The results are shown below. As $\lambda$ increases, the effect of the L2 regulations get stronger, resulting in performance degradations in all attacks. Nevertheless, YOQO still keeps comparable performance as the Boundary attack. Additionally we also notice that strong L2 regulation makes the training very unstable, and also causes performance degradations on the test accuracy.

## A.4 ADDITIONAL RESULTS ON BIGGER DATASETS AND PRETRAINED MODELS

We turn to use larger dataset or models pre-trained on large dataset to further strenthen the utility of the attack. Particularly, 1) we use larger training datasets with higher resolutions and more samples

| Regulation Param $\lambda$ | Gap Atk | YOQO Online | YOQO Offline | Boundary Atk |
|---|---|---|---|---|
| 0.01 | 60.10% | 72.34%% | 69.91% | 71.75% |
| 0.02 | 54.13% | 68.70% | 66.47% | 70.03% |
| 0.04 | 52.32% | 57.71% | 55.70% | 55.65% |

Table A.3: Studies on varying the weight decay of L2 regulations to minimize the overfitting.

(75k Tiny-ImageNet, 25k CIFAR-10); 2) we consider both training from scratch and fine-tuning a public model pre-trained on ImageNet. The inference accuracy of different attacks are shown in the Table A.4. We can observe that our attacks maintain comparable accuracy as SOTA attacks, but much more efficient (only query once). This conclusion is consistent with the previous evaluations in § 4, demonstrating the practicality and utility of our attacks in a wider set of configurations.

| Dataset | Model | Dataset Size | Gap | Boundary | Online | Offline |
|---|---|---|---|---|---|---|
| CIFAR10 | CNN7 from Scratch | 25k | 61.62% | 74.34% | 75.03% | 73.88% |
| CIFAR10 | ResNet34 Pretrained on Imagenet | 25k | 56.2% | 63.01% | 62.27% | 60.34% |
| Tiny-Imagenet | ResNet 34 Pretrained on Imagenet | 75k | 65.2% | 73.21% | 73.13% | 70.47% |
| Tiny-Imagenet | ResNet 34 From Scratch | 75k | 75.7% | 81.03% | 80.94% | 77.65% |

Table A.4: Membership inference accuracy of MIAs on bigger datasets and fine-tuned models.

## A.5 ADDITIONAL EXPERIMENTS ON TRANSFERABILITY

We conduct experiments using victim models trained with various settings that are different from the shadow models. Specifically, we test different optimizers, batch sizes, and learning rate. All the experiments are conducted using the same setting in the ablation studies. The results are shown in Table A.5.

| Optimizer of shadow models | Optimizer of target models | Batch Size | Learning Rate | Gap Atk | Online Atk | Offline Atk |
|---|---|---|---|---|---|---|
| Adam | SGD | 128 | 0.01 | 68.70% | 80.97% | 78.35% |
| Adam | SGD | 64 | 0.005 | 69.50% | 79.22% | 77.59% |
| Adam | AdamW | 128 | 0.01 | 67.50% | 80.01% | 77.43% |
| Adam | AdamW | 64 | 0.005 | 68.38% | 80.94% | 76.92% |

Table A.5: Transferability between different training settings.

We observe that the different training details such as batch size and optimizer have slight impacts on the effectiveness of our methods. The crafted query samples from YOQO have high transferability.

## A.6 COMPARISON TO METHOD IN WEN ET AL. (2022)

We noticed that our work shares a similar idea to that in (Wen et al. (2022)). Here we discuss the differences between our work and (Wen et al. (2022)) to justify our contribution.

The method proposed in (Wen et al. (2022)) is to work in the circumstances where the attacker is able to obtain the **logits of the model**, whereas our method is exclusively designed for the **label-only** situations. The difference in the threat model leads to the difference in the loss function design.

To further see the differences, we migrate the method in (Wen et al. (2022)) to the hard label scenario, and measure the attack effectiveness. We adopt the method in (Wen et al. (2022)) for the query-sample-generation (Algorithm 1 in our paper), while the rest parts are the same. We perform evaluations on CIFAR10 (2,500 samples) using CNN7. The rest of the settings are the same as the experiments in § 4. The results are shown in Table A.6 . It is clear that our methods are much more effective than (Wen et al. (2022)) in terms of label-only MIA.

| Loss term | Inference Accuracy |
|---|---|
| YOQO Online | 81.65% |
| Canary Online | 73.27% |
| YOQO Offline | 79.19% |
| Canary Offline | 71.54% |

Table A.6: Comparison between Canary and YOQO.

