# OpenReview forum: "You Only Query Once: An Efficient Label-Only Membership Inference Attack"
_ICLR.cc/2024/Conference — ICLR 2024 poster_

### Official Review · Reviewer_1W2d · 2023-10-22

**Soundness:** 3 good
**Presentation:** 3 good
**Contribution:** 2 fair
**Rating:** 6
**Confidence:** 3

**Summary:**

In this paper, the authors propose a novel label-only membership inference attack named YOQO. Before querying the target model, the attacker first perturbs the target image in such a way that it produces differing predictions between shadow models trained with and without the target image. Subsequently, the attacker queries the target model with the perturbed image and examines the predicted label. Remarkably, YOQO can achieve state-of-the-art performance with just a single query.

**Strengths:**

- The paper is well-written, making it easy to follow.
- The results are promising. The method requires only one query yet achieves results comparable to state-of-the-art methods.
- The extensive ablation studies presented in the paper are commendable, particularly the experiments under multiple defenses.
- The offline attack demonstrates impressive performance without necessitating the retraining of any shadow models, which is advantageous for practical applications.

**Weaknesses:**

While I'm impressed with the paper, a primary concern is the apparent similarity between the proposed method and the method in [1] from ICLR 2023. The latter utilizes a similar loss to craft queries, enhancing membership inference attacks. Their emphasis is on the threat model that extracts logits from the target model. However, [1] is absent from the paper's references.

[1] Wen, Y., Bansal, A., Kazemi, H., Borgnia, E., Goldblum, M., Geiping, J., & Goldstein, T. (2022). Canary in a Coalmine: Better Membership Inference with Ensembled Adversarial Queries. arXiv preprint arXiv:2210.10750.

**Questions:**

- As mentioned in the weaknesses section, what would be the main difference between the proposed method and the approach outlined in [1]?
- It's encouraging that the crafted query is transferable. However, if the training algorithm of the target model differs from that of the shadow model, such as having different learning rates or optimizers, would the attack still retain its potency?

---

> ### Author Response · Authors · 2023-11-19
>
> # To Reviewer 1W2d
> ## Q1: What would be the main difference between the proposed method and the approach outlined in [1]?
>
> We apologise for missing this important work in our paper. We will take your advice to cite and discuss this paper. Our attacks are highly different from this paper from the following perspectives.
>
> (1) **Difference in the Threat Models.**
> The method proposed in [1] is to work in the circumstances where the attacker is able to obtain the ***logits of the model***, whereas our method is exclusively designed for the ***label-only*** situations.
>
> (2) **Difference in the loss function.**
> The difference in the threat model also leads to the difference in the loss function design. In [1], since the attacker can get the predicted logits, the main purpose of the loss function is to ‘optimise $x$ so that IN shadow models have low losses on $x$ and OUT models have high losses on $x$’. It directly minimises the ***logits*** of the out models by using the loss term $-\log(1-f_\theta(x)_y)$ where $f_\theta(x)_y$ stands for the ***logits*** corresponding to the ground truth $y$, which will result in a smaller logits, i.e., smaller $f_\theta(x)_y$.
>
> In contrast, our attacker can only get the ***hard labels***. Therefore, the goal of our loss function is to optimise $x$ so that it can be correctly classified by IN models while being mis-classified by the OUT models. In this case, it is problematic to just enlarge the variance in the logits, confidence scores or the loss values, as it may not be enough to cause the differences in the predicted labels of the query samples between the in and out models. To solve this problem, we propose to use $CE(f_{out}^i(x),l_i')$, where $l_i'$ is the 'nearest' class of the target sample $x$ other than its ground truth. Unlike $-\log(1-f_\theta(x)_y)$, the $CE(f_{out}^i(x),l_i')$ term tends to ensure the ***label difference***. When minimising the cross-entropy, it enlarges the logits of target label l_i’ at the mean time minimising the other logits to guarantee the predicted-label-level differences. Moreover, to carefully choose the target label $l_i’$ helps us to push the query sample into the ‘improvement area’, i.e. the most sensitive part of the decision boundary towards the membership difference between the in and out models, therefore further enhancing the attack. (We’ve demonstrated the effectiveness of choosing target labels in the ablation study of the manual.)
>
> (3) **Difference in the attack effectiveness**
> To further see the differences, we migrate the method in [1] to the hard label scenario, and measure the attack effectiveness. We adopt the method in [1] for the query-sample-generation (Algorithm 1 in our paper), while the rest parts are the same. We perform evaluations on CIFAR10 (2,500 samples) using CNN7. The rest of the settings are the same as the experiments in the paper. The results are shown in the table below. It is clear that our methods are much more effective than [1] in terms of label-only MIA.
>
> Loss term|Inference Accuracy
> ---|---
> YOQO Online| 81.65%
> Canary[1] Online| 73.27%
> YOQO Offline| 79.19%
> Canary[1] Offline| 71.54%
>
> # Q2: It's encouraging that the crafted query is transferable. However, if the training algorithm of the target model differs from that of the shadow model, such as having different learning rates or optimizers, would the attack still retain its potency?
>
> We conduct experiments using victim models trained with various settings that are different from the shadow models. Specifically, we test different optimizers, batch sizes, and learning rate. All the experiments are conducted using the same setting in the ablation studies. The results are shown below.
>
> Optimizer of shadow models|Optimizer of target models|Batch Size|Learning Rate|Gap Attack|Online Attack|Offline Attack
> ---|---|---|---|---|---|---
> Adam|SGD|128|0.01|68.70%| 80.97% | 78.35%
> Adam|SGD |64 |0.005|69.05%|79.23%|77.59%
> Adam|AdamW|128|0.01 |67.50%| 80.01% |77.43%
> Adam|AdamW |64 |0.005|68.38%|79.31%|76.92%
>
> We observe that the different training details such as batch size and optimizer have slight impacts on the effectiveness of our methods. ***The crafted query samples from YOQO have high transferability.***
>
> ## Reference:
> [1] Wen, Y., Bansal, A., Kazemi, H., Borgnia, E., Goldblum, M., Geiping, J., & Goldstein, T. (2022). Canary in a Coalmine: Better Membership Inference with Ensembled Adversarial Queries. arXiv preprint arXiv:2210.10750.

---

> > ### Comment · Reviewer_1W2d · 2023-11-19
> > **Thanks for the response.**
> >
> > I greatly appreciate the authors' detailed explanations and additional experiments. The proposed method is clearly superior to [1], though they share a similar concept. Moreover, its robustness to various training algorithms further solidifies the work. Therefore, I have raised my rating score accordingly.

---

> > > ### Author Response · Authors · 2023-11-19
> > >
> > > Thank you for your helpful advice to make our work more solid and rigorous. Please let us know if you have more concerns.

---

### Official Review · Reviewer_1cwK · 2023-10-26

**Soundness:** 3 good
**Presentation:** 3 good
**Contribution:** 3 good
**Rating:** 8
**Confidence:** 4

**Summary:**

The paper introduces YOQO, a novel label-only membership inference attack designed to address privacy threats in machine learning models. YOQO's key innovation lies in its identification of an "improvement area" around a target sample, enabling the crafting of query samples with hard labels that effectively determine the target sample's membership, significantly reducing the query budget required from over 1,000 queries to just one query. The study demonstrates that YOQO exhibits effectiveness comparable to state-of-the-art Membership Inference Attacks (MIA) while demonstrating greater resilience against various defense mechanisms. This underscores its significance in the context of privacy attacks on machine learning models.

**Strengths:**

- YOQO introduces a label-only membership inference attack, reducing the query budget from over 1,000 queries to just one query.
- The attack demonstrates effectiveness on par with state-of-the-art MIA methods, indicating its practical relevance.
- YOQO exhibits greater robustness against various defense mechanisms, underlining its potential for real-world applications.

**Weaknesses:**

- The use of accuracy as the primary evaluation metric is questioned, as it may not reflect worst-case performance in MIA.
- The potential for overfitting in the experimental setting due to a small training dataset (2,500 samples) raises concerns about the generalizability of the results.
- While the transformation of Equation 1 into a minimization problem is justified, the exploration of alternative techniques, such as introducing a weight term, could enhance the paper's depth and robustness.

**Questions:**

Two noteworthy concerns arise in the evaluation of the paper.

First, in terms of the chosen evaluation metric, the authors have opted for accuracy to assess the performance of their attack. However, there are concerns about the appropriateness of this choice. While I agree that using a log-scale ROC curve may not be practical for this case, the fundamental issue at hand pertains to the need for evaluating the worst-case scenario for MIA. Based on my understanding, samples closer to the decision boundary should have a higher probability of being classified as members, while those significantly distant from this boundary may exhibit a lower likelihood of membership. Consequently, assessing accuracy across all samples may not effectively unveil the worst-case performance. To address this, a suggestion is made to focus on accuracy calculations for samples located farthest from the decision boundary, even though the attacker may lack knowledge of the specific location of such samples. The presence of ground truth information during the evaluation phase supports the feasibility of this approach.

Second, the experimental setting raises concerns regarding overfitting. The target model is trained with a relatively small dataset of 2,500 samples, which could potentially result in severe overfitting. Although the paper does not explicitly state the training and testing accuracy gap for the target model, it can be inferred from the GAP attack that the overfitting level exceeds 40%, potentially impacting the reliability of the conclusions. I would like to see the authors conduct experiments on well-generalized models, involving a larger training dataset to mitigate the risk of overfitting.

In Section 3.2, when the authors optimize the x to get the improvement area, they believe that Equation 3 won’t work since scalars in both in models and out models are in [0,1]. I agree, and to address this constraint, the authors adapt Equation 1 into a minimization problem, which is a common practice in finding adversarial examples. Just out of curiosity, I wonder whether adding a weight term $\lambda$ to Equation 2 could achieve the same result, that is, turn the output of in models to [0,$\lambda$].

---

> ### Author Response · Authors · 2023-11-19
>
> # To Reviewer 1cwK
> ## Q1: The use of accuracy as the primary evaluation metric is questioned, as it may not reflect worst-case performance in MIA.
>
> We appreciate your suggestions on how to show the worst-case performance in MIA, which is really inspiring. Following your suggestions, we use the distance provided by boundary attack [1] as the metric to pick the worst-case samples, which directly measures the distance (L0, L1, L2, L $\infty$) in the space of the target samples. W e calculate the distance on 5 victim models and take the averages as the distance. Then we select the top 10% samples (200 samples) with the largest distance. The results are shown below:
>
> Model Type|L0|L1|L2|L $\infty$
> ---|---|---|---|---
> Gap Attack |50.0%|50.0%|50.0%|50.0%
> Boundary Accuracy| 52.7% | 54.0% | 54.1% | 53.3%
> Online Accuracy | 65.3% | 61.0% | 61.22% | 61.5%
> Offline Accuracy| 60.0% | 57.7% | 56.5% | 59.5%
>
> We believe this result can reflect the worst-case situation to some extent. We observe that ***our methods tend to have better performance than the boundary attack. This indicates that YOQO can be more effective to infer the membership of the worst-case samples which the boundary attack is hard to deal with.*** To strengthen the solicity of our work, we will add the discussions in the revision.
>
> ## Q2:  The target model is trained with a relatively small dataset of 2,500 samples, which could potentially result in severe overfitting. I would like to see the authors conduct experiments on well-generalised models, involving a larger training dataset to mitigate the risk of overfitting.
>
> (1)We would like to clarify that ***we varied the size of the training dataset from 1,500 to 10,000 in the paper to show how the overfitting would affect the attack performance.*** The small dataset of 2,500 samples is only used for defence evaluation. As shown in Fig.2, the smallest gap between the test and train set is around 24% (training set accuracy = 98%, test set accuracy = 74%)
>
> (2)Following your suggestions, we further do the experiments using bigger datasets (25,000 samples of CIFAR10 and 75,000 samples of Tiny-ImageNet) to reduce the overfitting. To make the target model more well-generalised, we also conduct experiments with models pretrained on Imagenet. The results are shown in the table below. ***In all cases, YOQO retains its comparable performance as the boundary attack.***
>
> Dataset|Datasize|Model Type|Gap Attack|Boundary Attack|YOQO Online|YOQO Offline
> ---|---|---|---|---|---|---
> CIFAR-10|25,000|CNN7 from Scratch|61.62%| 74.34% | 75.03%|73.88%
> CIFAR-10|25,000|ResNet34 Pretrained on      Imagenet|56.2%|63.01%|62.27%|60.34%
> Tiny-Imagenet|75,000|ResNet34 Pretrained on Imagenet|65.2%|73.21%|73.15%|70.47%
> Tiny-Imagenet|75,000|ResNet34 From Scratch|75.7%|81.03%|80.94%|77.65%
>
> (3)Additionally, to explore the impact of overfitting on attack effectiveness, we conduct the experiments using L2 regulation to further reduce the overfitting. The results are shown below. ***Our attacks are still comparable with SOTA attacks but with much more efficiency (query only once).***
>
> Regulation Param $\lambda$|Gap Attack|YOQO Online|YOQO Offline|Boundary Attack
> ---|---|---|---|---
> 0.01| 60.10% | 72.32% | 69.91% | 71.75%
> 0.02| 54.13% | 68.70% | 66.47% | 70.03%
> 0.04| 52.32% | 57.71% | 55.70% | 55.65%
>
> We will add all the discussion and evaluation results of overfitting in the revised paper.
>
> ## Q3: Whether adding a weight term to Equation 2 could achieve the same result, that is, turn the output of in models to [0, $\lambda$].
>
> Indeed, having a weight term $\lambda$ can scale the output of in models to [0, $\lambda$], which may ameliorate the imbalance between the two terms. However, the real problem lies in the range of $CE(\frac{\lambda \cdot f_{in}}{f_{out}})$. For any given $f_{in}$, the loss $CE(\frac{\lambda \cdot f_{in}}{f_{out}}, l)$ ranges in $(-\infty, CE(\lambda \cdot f_{in}, l)]$, which means for any threshold $t$, we can theoretically find a $f_{out}$, so that $CE(\frac{\lambda \cdot f_{in}}{f_{out}})<t$, indicating the algorithm is not convergent. This means we cannot simply set a threshold on the total loss to decide when the algorithm should stop. In this case, we need to carefully choose the total iterations of the optimization as well as the $\lambda$ to ensure the acceptable inference accuracy, which is not that practical in the real word setting, and also means it is a sub-optimal choice (for the best $\lambda$ and iteration numbers can vary from samples to samples). Below we do the experiments to demonstrate the our opinion. We varies the $\lambda$ from 1 to 16. The results indicate that the performance in this case is more dependent on the $\lambda$, and is less effective in comparison with the original loss as in Fig.6 shows in the paper, which used the same settings of this experiment, i.e., 2,500 samples on CIFAR10.
>
> $\lambda$|1|2|4|8|16
> ---|---|---|---|---|---
> Inference Accuracy|72.13%|74.15%|75.25%|75.21%|75.07%

---

> > ### Comment · Reviewer_1cwK · 2023-11-21
> >
> > Thank you for the response! My concerns are adequately addressed, therefore I raise my score from 6 to 8.

---

> ### Author Response · Authors · 2023-11-22
>
> Thank you for the suggestions to enhance the robustness and rigor of our work. Please feel free to reach out if any further concerns arise.

---

### Official Review · Reviewer_bo89 · 2023-10-30

**Soundness:** 2 fair
**Presentation:** 3 good
**Contribution:** 3 good
**Rating:** 6
**Confidence:** 4

**Summary:**

The paper proposes YOQO a label-only MIA which, unlike prior SOTA label-only MIAs, queries the target model only once using a specific sample x’ derived from the target sample (x, l) with the goal of determining the membership of (x, l). YOQO finds the query sample x’ in the improvement region which is the difference in the decision boundaries due to insertion of (x, l) in the training data. YOQO proposes an optimization to find x’ that maximizes the error of OUT models on x’ and minimize the error of IN models on x’, and solves it using gradient decent. Evaluations show that YOQO is as effective or better than the existing label only MIAs on multiple datasets/model architectures.

**Strengths:**

- YOQO is a novel idea that reduces query budget required for MI
- Idea is well explained and paper is easy to read.
- Experiments are well designed to demonstrate the efficacy of the attack

**Weaknesses:**

- YOQO evaluations consider very small datasets and training data sizes
- Models might be overfitted on training data
- Unclear how this attack will work in practical settings

**Questions:**

This paper proposes a very novel YOQO attack and I think the paper also does a good job of explaining and evaluating the attack. The evaluations clearly show that YOQO is the new SOTA label only MIA. I have the following few concerns about the paper:

- Although the experimental setup considered is the common setup in most prior works, it contains mostly small datasets with very small training dataset sizes. Can authors perform experiments on larger datasets, e.g., Imagenet?
- I could not figure out the training recipe from the main paper. In particular does the training ensure that the models don’t overfit to the training data, e.g., using L2 regularization or any other common regularization techniques? I feel the criterion that training goes on until a model has >98% accuracy on training data may lead to overfitting that will lead to unnecessarily stronger MIAs.
- On the same lines as the above comment, can authors add a few more details of various dataset sizes used in training with/without defenses, e.g., adversarial regularization? Also can they add details of training procedure? These are important given that MIA efficacy is greatly affected by these factors.
- Utility of the attack: I could not understand from the current evaluations how the attack will perform in the real world. The dataset sizes are quite small which is seldom the case now a days, unless if model is fine-tuned using a small dataset. Can authors provide results for some real-world settings? Some suggestions: 1) use larger datasets 2) fine-tune a model pre-trained on large datasets?

---

> ### Author Response · Authors · 2023-11-19
>
> # To Reviewer bo89: (1/2)
> ## Q1: Although the experimental setup considered is the common setup in most prior works, it contains mostly small datasets with very small training dataset sizes. Can authors perform experiments on larger datasets, e.g., Imagenet?
>
> Response:
>
> Thanks for the great suggestion. ***The size of the training dataset used in the current manuscript ranges from 1,500 to 10,000, with various data types (e.g., images, tables). These are indeed common setups in existing MIA works.*** Following your suggestion, we'd like to extend the evaluation to larger datasets. Performing evaluations on Imagenet requires quite a long time, especially the online attack, which will exceed the response deadline. Alternatively, we conduct extra experiments on Tiny-Imagenet, a subset of Imagenet which contains over 100k images of 64 $\times$ 64 size belonging to 200 categories. The evaluation results can be found in our response to your Q4. Our method remains effective in larger datasets. The results can be found in the second part of the response.
>
> ## Q2: I could not figure out the training recipe from the main paper. In particular, does the training ensure that the models don’t overfit to the training data, e.g., using L2 regulation or any other common regulation techniques? I feel the criterion that training goes on until a model has >98% accuracy on training data may lead to overfitting that will lead to unnecessarily stronger MIAs.
>
> Response:
> (1)When training the models, we indeed used the L2 regulations and set the regulation parameter $\lambda$ to be 5e-04. The influence of L2 regulation may not be strong enough to entirely prevent overfitting. Yet it is a common setting for training models on the datasets we use, and we believe the overfitting is reasonable, the same as other MIA works.
>
> To demonstrate the effectiveness of YOQO on less overfitting models, we further perform evaluations on models trained with stronger L2 regulations. Specifically, we use CNN7 as both the shadow and victim model, over 2,500 CIFAR-10 images as the training set. The results are shown below. As the weight decay $\lambda$ increases, the effect of the L2 regulations get stronger, resulting in  performance degradations in all of the attacks. Nevertheless, ***YOQO still keeps comparable performance as the Boundary attack.*** Additionally we also notice that strong L2 regulation makes the training very unstable, and also causes performance degradations on the test accuracy.
>
> Regulation Param $\lambda$|Gap Attack|YOQO Online|YOQO Offline|Boundary Attack
> ---|---|---|---|---
> 0.01| 60.10% | 72.32% | 69.91% | 71.75%
> 0.02| 54.13% | 68.70% | 66.47% | 70.03%
> 0.04| 52.32% | 57.71% | 55.70% | 55.65%
>
> (2) For the condition of training termination, we would like to clarify that ***we did not take the training set accuracy as the criterion to stop training.*** Instead, for each task and dataset, ***we stop the training when the model achieves the best performance on the validation set.*** Such models will also get >98% accuracy on the training set. Sorry for the confusion, and we will add all the training details in the revised paper.
>
> (3)Additionally, the models evaluated in our paper are actually less overfitting in comparison with the Boundary attack paper [1]. For instance, using 2,500 samples from CIFAR10 for training, the ASR of gap attack in our work is around 70%, while that of [1] is around 75% (Fig.1 of [1]). As the gap attack is correlated to the gap between the performances on the train and test sets, a lower ASR in gap attack means our models are less overfitting in comparison with that being tested in the Boundary attack paper [1].
>
> ## Q3: On the same lines as the above comment, can authors add a few more details of various dataset sizes used in training with/without defences, e.g., adversarial regulation? Also can they add details of training procedure? These are important given that MIA efficacy is greatly affected by these factors.
>
> Response:
> Thank you for pointing this out. We are sorry for missing the training details due to the page limit. In our paper, ***for all the defence evaluations, we use 2,500 CIFAR-10 samples as the training dataset. For the attack effectiveness evaluations, we use different datasets with different numbers of samples from 1,500 to 10,000.***  These details can be found at the beginning of Sections 4.2 and 4.3. Our training settings are shown below:
>
> Dataset|Model|Learning Rate|Optimizer|Batch Size|Training Epoch
> ---|---|---|---|---|---
> CIFAR10/CIFAR100/gtsrb/svhn|CNN7|0.001|Adam|128|30
> CIFAR10/CIFAR100/gtsrb/svhn|VGG16|0.001|Adam|64|50
> CIFAR10/CIFAR100/gtsrb/svhn|ResNet18|0.001|Adam|64|20
> CIFAR10/CIFAR100/gtsrb/svhn|DenseNet121|0.001|Adam|128|30
> CIFAR10/CIFAR100/gtsrb/svhn|InceptionV3|0.001|Adam|128|25
> CIFAR10/CIFAR100/gtsrb/svhn|SeResNet18|0.001|Adam|128|20
> Purchase100/location|ColumnFC|0.001|Adam|32|30
>
> We will clarify those details in the revised manuscript.

---

> ### Author Response · Authors · 2023-11-19
>
> # To Reviewer bo89: (2/2)
>
> ## Q4: Utility of the attack: I could not understand from the current evaluations how the attack will perform in the real world. The dataset sizes are quite small which is seldom the case nowadays, unless the model is fine-tuned using a small dataset. Can authors provide results for some real-world settings? Some suggestions: 1) use larger datasets 2) fine-tune a model pre-trained on large datasets?
>
>
> Thank you for the suggestions. We follow them to evaluate our attacks on these new settings. Particularly, 1) we use larger training datasets with higher resolutions and more samples (75k Tiny-ImageNet, 25k CIFAR-10); 2) we consider both training from scratch and fine-tuning a public model pre-trained on ImageNet. The inference accuracy of different attacks are shown in the following table. We can observe that our attacks maintain comparable accuracy as SOTA attacks, but much more efficient (only query once). This conclusion is consistent with the evaluations in the paper, demonstrating the practicality and utility of our attacks in a wider set of configurations.
>
> Dataset|Datasize|Model Type|Gap Attack|Boundary Attack|YOQO Online|YOQO Offline
> ---|---|---|---|---|---|---
> CIFAR-10|25,000|CNN7 from Scratch|61.62%| 74.34% | 75.03%|73.88%
> CIFAR-10|25,000|ResNet34 Pretrained on      Imagenet|56.2%|63.01%|62.27%|60.34%
> Tiny-Imagenet|75,000|ResNet34 Pretrained on Imagenet|65.2%|73.21%|73.15%|70.47%
> Tiny-Imagenet|75,000|ResNet34 From Scratch|75.7%|81.03%|80.94%|77.65%
>
>
>
> ## References:
>
> [1] Christopher A Choquette-Choo, Florian Tramer, Nicholas Carlini, and Nicolas Papernot. Label-only membership inference attacks. In International conference on machine learning, pp. 1964–1974.
> PMLR, 2021.

---

> > ### Comment · Reviewer_bo89 · 2023-11-21
> > **Thanks for the response!**
> >
> > Thanks for the detailed response and new results. I have a few more questions:
> >
> > - in [1], Figure 2-c, I see that for cifar10 trained on 2500 training samples, boundary distance attack with random noise based augmented samples achieves close to 80% accuracy with just 10 queries. Why are your results so different?
> >
> > - current threat model assumes that the attacker has data from the exact same distribution as the training data of the target model. I feel this is quite a strong assumption that even some of the previous works make. Can you comment on how practical this is and how would the attack perform if the attacker does not have such access? I think it is important to understand how these attacks will work in practice.
> >
> > - in the 'Hyper-parameter in loss functions' section: it says that with increasing alpha, there are fewer FPs. If i understand correctly, this means you can keep increasing \alpha to get low FPRs and measure TPRs at low FPRs?
> >
> > - I guess the idea here is to query the target model only once to infer membership and looks like the attack does close to boundary attack. But, if the query budget is higher, e.g., 10 queries, will this attack be able to surpass boundary attack, e.g., in case of Purchase100? How would attacker design those queries?
> >
> > - i see that offline attack many times outperforms the online attack; why is this the case?
> >
> > Comments:
> > - Add model archs in Table 3 and training data size in caption of table 2.

---

> > > ### Author Response · Authors · 2023-11-22
> > > **Response to the new questions to Reviewer bo89 part (1/2)**
> > >
> > > Thank you for your comment and questions. Below we give the response
> > > ***Questions:***
> > >
> > > > Why are your results so different from that in Figure 2-c from [1]?
> > >
> > > The key reason is that ***the models evaluated in [1] are more overfitting to the dataset.*** We can see from Fig.2-c in [1] that although the queries augmented by random noise are able to achieve around 80% ASR, the ASR of gap attack is also 75%, indicating the test set accuracy is 50% less than the training set accuracy. This means the models in [1] are more overfitting to cause stronger MIA. In contrast, the models in our paper are less overfitting (the ASR of gap attack is around 68%), making our evaluations more practical.
> > > Therefore, both the boundary distance attack by random noise or adversarial queries in [1] seems to have better performance.
> > >
> > > > Can you comment on how practical this is and how would the attack perform if the attacker does not have such access (gaining similar distribution)?
> > >
> > > This assumption is widely accepted in many adversarial machine learning papers, not only MIA [1, 2, 3], but also model extraction attacks [4, 5], black-box adversarial attacks [6, 7], watermark removal attacks [8], and others. In practice, it is easy for the attacker to obtain a dataset following the same distribution as the victim’s training set. The victim models are normally trained on the data of the distribution close to the real world. So the attacker can collect similar data by sampling from the real world. Besides, as our method does not quite rely on the exact size of the dataset (Figure 5), this shadow dataset can be much smaller than the victim’s training set, making it easier for the attacker to collect.
> > >
> > > In case the attacker does not have such access, he can adopt some pre-trained models which are trained from universal real-world datasets, and then fine-tune it with the target samples to get the in or out models, as shown in the new experiments we add in the revised manuscript.
> > >
> > >
> > > > If I understand correctly, this means you can keep increasing $\alpha$ to get low FPRs and measure TPRs at low FPRs?
> > >
> > > Although we can minimise FPRs by increasing $\alpha$, the effect (especially when it is bigger than 2) becomes marginal when it goes bigger. We can see from Figure 6-(a) that the effect of adjusting $\alpha$ becomes smaller and smaller, making it hard to further adjust the FPRs.
> > >
> > > > If the query budget is higher, e.g., 10 queries, will this attack be able to surpass boundary attack, e.g., in case of Purchase100? How would attacker design those queries?
> > >
> > > We can actually design those queries by just adding small random noises to the initial stage of the generation process depicted in Algorithm 1. Then we can query the models and decide the membership by a majority vote according to the label we get. However, we are not sure how this would affect the inference accuracy. Possibly it may decrease the FPRs and also cause degradation to the TPRs. Indeed this is a promising direction, and we think a more sophisticated approach is needed to surpass the boundary attack, e.g., to design several correlated query samples instead of just adding random noises at the initial stage. We will consider this as future work.
> > >
> > > > I see that offline attack many times outperforms the online attack; why is this the case?
> > >
> > > In the evaluation of the paper, we kept all the parameters the same for all the experiments except for the ablation study, which means they may not be optimal for all the models and tasks being evaluated. We think the phenomenon that the offline attack sometimes performs better is because we happened to obtain a set of hyper-parameters that are more suitable for the tasks and models. This, however, doesn't mean offline attack is superior to the online attack in these tasks.
> > >
> > > ***Comments***
> > >
> > > > Add model archs in Table 3 and training data size in caption of table 2.
> > >
> > > Thank you again for the suggestion! We have updated the manuscript according to your advice.

---

> > > > ### Author Response · Authors · 2023-11-22
> > > > **Response to the new questions to Reviewer bo89 part (2/2)**
> > > >
> > > > References
> > > >
> > > > [1] Christopher A Choquette-Choo, Florian Tramer, Nicholas Carlini, and Nicolas Papernot. Label-only membership inference attacks. In International conference on machine learning, pp. 1964–1974. PMLR, 2021.
> > > >
> > > > [2] Nicholas Carlini, Steve Chien, Milad Nasr, Shuang Song, Andreas Terzis, and Florian Tramer. Membership inference attacks from first principles. In 2022 IEEE Symposium on Security and Privacy (SP), pp. 1897–1914. IEEE, 2022
> > > >
> > > > [3] Wen, Y., Bansal, A., Kazemi, H., Borgnia, E., Goldblum, M., Geiping, J., & Goldstein, T. (2022). Canary in a Coalmine: Better Membership Inference with Ensembled Adversarial Queries. arXiv preprint arXiv:2210.10750.
> > > >
> > > > [4] Guanlin Li, Guowen Xu, Shangwei Guo, Han Qiu, Jiwei Li, and Tianwei Zhang. Extracting robust models with uncertain examples. In The Eleventh International Conference on Learning Representations, 2022
> > > >
> > > > [5] Papernot, N., McDaniel, P., Goodfellow, I., Jha, S., Celik, Z. B., & Swami, A.(2017, April). Practical black-box attacks against machine learning. In Proceedings of the 2017 ACM on Asia conference on computer and communications security (pp. 506-519).
> > > >
> > > > [6] Nathan Inkawhich, Kevin J Liang, Lawrence Carin, and Yiran Chen. Transferable perturbations of deep feature distributions. ICLR, 2020.
> > > >
> > > > [7] Li, M., Deng, C., Li, T., Yan, J., Gao, X., & Huang, H. (2020). Towards transferable targeted attack. In Proceedings of the IEEE/CVF Conference on Computer Vision and Pattern Recognition (pp. 641-649).
> > > >
> > > > [8] Wu, Dongxian, and Yisen Wang. "Adversarial neuron pruning purifies backdoored deep models." Advances in Neural Information Processing Systems 34 (2021): 16913-16925.

---

### Author Response · Authors · 2023-11-21

We thank all the reviewers for your valuable feedback!

In response to your insights and suggestions, we have revised our manuscript. Modifications in the updated manuscript are highlighted in blue for easy identification. Below we specify the amendments:

- 1W2d: (Section 2.1) Adding discussion and reference about the mentioned work [1]


- 1cwK, bo89: (Section 4) Adding details for newly evaluated models and datasets.

- 1cwK, bo89: (Section 4.1) Attack performance on Tiny-ImageNet and pre-trained models.

- bo89:(Appendix A.1) Adding training recipes of the models.

- 1cwK:(Appendix A.2) Adding the results of some worst-case studies.

- bo89:(Appendix A.3) Adding the results of studies on L2 regulation.

- bo89:(Appendix A.4) Adding YOQO performance on pretrained models and larger dataset.

- 1W2d:(Appendix A.5) More results for transferability evaluation (between different optimizer and learning rate).

- 1W2d:(Appendix A.6) Comparison to the mentioned work [1].

We hope these improvements and our responses can address your concern. We sincerely appreciate your guidance and support.

[1] Wen, Y., Bansal, A., Kazemi, H., Borgnia, E., Goldblum, M., Geiping, J., & Goldstein, T. (2022). Canary in a Coalmine: Better Membership Inference with Ensembled Adversarial Queries. arXiv preprint arXiv:2210.10750.

---

> ### Author Response · Authors · 2023-11-22
> **More updates in the manuscript**
>
> We thank all the reviewers again for your valuable feedback!
>
> We further adjust the manuscript following the suggestions of Reviewer bo89.
>
> - section (4.2 table.3): adding model architectures.
>
> - section (4.2 table.2): adding the size of dataset in the caption.
>
> all the parts that are different from the original manusacript are in blue for identifications.
>
> We hope these improvements and our responses can address your concern. We sincerely appreciate your guidance and support

---

### Meta-Review · Area_Chair_n2n8 · 2023-12-07

**Metareview:**

This paper proposes a new label-only membership inference attack that only requires a single query to the target model. Authors evaluated the attack on both image and tabular datasets, using a variety of model architectures, and showed that it performs well compare to prior work while requiring much fewer label-only queries. Reviewers unanimously agree that the idea presented is novel, effective, and advances the state-of-the-art.

One common criticism is that the evaluated datasets are relatively small scale, and thus it is unclear if the method will be practical for real world applications. The primary performance metric, namely MIA accuracy, also deviates from the current best practice of using AUC and TPR @small FPR to identify the true efficacy of membership inference on vulnerable samples. The authors are strongly encouraged to address these weaknesses in the camera ready version.

**Justification For Why Not Higher Score:**

The weaknesses of using small scale dataset and using MIA accuracy as the performance metric raise questions about the practicality of the attack. I believe the paper is still of great value due to its novel idea, but more solid evaluation is needed to prove its strong empirical performance with certainty.

**Justification For Why Not Lower Score:**

The idea is novel and its empirical performance is encouraging based on relatively small scale experiments. Reviewers unanimously voted for acceptance.

---

### Decision · Program_Chairs · 2024-01-16

Accept (poster)